# Epidemiology of Adverse Outcomes in Teenage Pregnancy—A Northeastern Romanian Tertiary Referral Center

**DOI:** 10.3390/ijerph20021226

**Published:** 2023-01-10

**Authors:** Alexandra Ursache, Ludmila Lozneanu, Iuliana Bujor, Alexandra Cristofor, Ioana Popescu, Roxana Gireada, Cristina Elena Mandici, Marcel Alexandru Găină, Mihaela Grigore, Daniela Roxana Matasariu

**Affiliations:** 1Department of Obstetrics and Gynecology, University of Medicine and Pharmacy “Grigore T. Popa”, 700115 Iasi, Romania; 2Department of Obstetrics and Gynecology, Cuza Vodă Hospital, 700038 Iasi, Romania; 3Department of Morpho-Functional Sciences I—Histology, Pathology, University of Medicine and Pharmacy “Grigore T. Popa”, 700115 Iasi, Romania; 4Department of Dermatology, University of Medicine and Pharmacy “Grigore T. Popa”, 700115 Iasi, Romania; 5Psychiatry, Department of Medicine III, University of Medicine and Pharmacy “Grigore T. Popa”, 700115 Iasi, Romania

**Keywords:** childhood, underage pregnancy, teenage pregnancy, quality of life, girl pregnancy, education, healthcare system, neonatal outcomes

## Abstract

Background: Despite being a very well-documented subject in the literature, there are still conflicting results regarding teenage pregnancies and their fetal outcomes. Methods: We conducted a retrospective, comparative cohort study that included 1082 mothers aged less than 18 years, compared to 41,998 mothers aged over 18 years, who delivered in our tertiary referral center between January 2015 and December 2021. To check for significant differences between the two groups, the chi-qquared or Fisher’s test for categorical variables were used. Results: We detected statistically significant higher rates of fetal malformation, premature birth, FGR and SGA fetal growth conditions, preeclampsia, condylomatosis and vaginal infection with E. coli in our cohort of teenagers. In this subpopulation of teenagers, the rate for premature birth at less than 32 weeks of gestation was 3.26-fold higher and 3.25-fold higher for condylomatosis, and these results referred to the cohort of adult patients (>18 years old) that gave birth in the same interval of time. Conclusions: Teenage pregnancies still remain a major health problem that burdens all countries worldwide regardless of their income. It needs solutions initially to prevent pregnancy in this young age segment and last but not least to improve both maternal and fetal outcomes.

## 1. Introduction

The transitional stage of physical and psychological development from childhood to adulthood, comprising the totality of changes in social environment, mental and biological health, defines “adolescence”. The World Health Organization (WHO) describes an adolescent or a teenager as those between 10 and 19 years of age [1]. It is a critical stage of development during which any stressful experience may deleteriously impact stages from adolescent development to healthy adulthood. A pregnancy can be such a stressful experience, as teenagers are not physically and mentally mature enough to experience challenges [2].

Pregnancy in adolescence is defined as teenage pregnancy between 10 and 19 years old. It is a proven fact that a pregnancy in this stage of a woman’s life involves an increased risk of adverse reproductive outcomes. It remains unclear whether this is mainly attributable to adverse sociodemographic factors, social inequalities or just to the biological immaturity of adolescents [3].

About 11% of births worldwide involve teenage women, and unfavorable pregnancy outcomes account for 23% of the overall burden of disease in girls aged 15–19 years. Over 90% of adolescent pregnancy complications are recorded in developing countries [4]. WHO 2018 statistics show that about 16 million girls aged 15 to 19 years old and approximately 1 million under the age of 15 become pregnant and give birth every year. Most are from low- or middle-income countries around the world [1,2].

Over the past 15 years, most countries reported a continuous decline in teenage pregnancy rate, attributable to teenage support, education, contraception and other pregnancy prevention strategy [5]. The unfavorable outcomes of teen pregnancy are important public health problems with major socioeconomic impacts [6]. Although favorable pregnancy outcomes are known to be less common in teenagers than in older women, the real cause of these complications, biological or socioeconomic, remains uncertain. Teen pregnancy is becoming more and more a public health issue than a clinical practice problem [7]. The vast majority of teenage pregnancies are unintended and unplanned [8].

Teenage pregnancies are at higher risk of a wide range of life-threatening complications with serious health sequelae in later life. Very young mothers are at increased risk for complications such as preeclampsia and eclampsia, anemia, puerperal endometritis, hemorrhage, chronic infection and also socioeconomic deprivation. Teenage pregnancy is also associated with risks for adverse delivery outcomes: stillbirth, preterm birth and low birth weight [9,10,11].

Teenagers seem to have insufficient gestational weight gain resulting in low birthweight infants and intergenerational malnutrition later. It seems that the neonatal mortality rate in teenage pregnancies is about 3-fold higher than in adult women, and the risk of maternal deaths due to medical complications is 2-fold higher [12].

There are many manifest differences in teenage pregnancy rates and outcomes between states and within a state and between different demographic regions [13]. Studies on teen pregnancy rates in Europe showed higher rates in the eastern regions. A high incidence of teenage pregnancies was found in Bulgaria and Romania, a moderate incidence in Slovakia and a low incidence in the remaining European countries. Teenage pregnancy rates in Eastern Europe countries were 2- to 3-fold higher than in other European countries [14].

Despite being a very well-documented subject that is always in the center of attention, in the literature, there are still conflicting results regarding teenage pregnancies and their fetal outcomes. These might be due to different study settings, design, sample size, socioeconomic aspects, the quality of healthcare systems and of course healthcare access [15]. 

The few studies on this topic in Romania demonstrated that it is still a burden on our healthcare system, and accurate and reliable information remain unavailable. The eastern regions are deemed to be the least developed.

The aim of our study was to determine the pregnancy rate in underage adolescents and identify the factors impacting the pregnancy outcomes. In our country, a teenager is considered to be those less than 18 years old. 

In this article, four areas are reviewed: (1) trends in teenage birth rates over time; (2) recurrent teenage pregnancy; (3) abortion rates; (4) associated disorders and disparities between urban and rural teenagers in this respect.

## 2. Patients and Methods

This pediatric retrospective cohort study included all mothers aged less than 18 years who delivered in our tertiary referral center, the Iasi “Cuza Voda” University Hospital, between January 2015 and December 2021. Included in the study were teenagers with a gestational age at delivery of more than 24 gestation weeks and a fetal birth weight of over 500 g. We enrolled 1082 underage pregnant adolescents and 41,998 pregnant women over 18 years old. 

The data regarding the young mothers were selected from the hospital’s electronic health records. The gestational age was calculated as the interval between the date of delivery and the date of the last normal menstrual period, and it was adjusted by using either first or second trimester morphological scans or the only ultrasound assessment the patient underwent. The data were checked for validity or missing values and then corrected by reviewing both the mother and newborn’s medical records.

All data analyzed in our statistics (blood tests, vaginal and urinary probes for culture) were obtained upon admission to our hospital.

The diagnostic criteria for the associated disorders were as follows:-Maternal anemia—a maternal hemoglobin concentration less than 11 mg/dL [16];-Fetal growth restriction (FGR)—an estimated fetal weight at or below the 3rd percentile compared to normal fetal weight for gestational age associated with abnormal Doppler [17];-Fetuses small for gestational age (SGA)—fetuses with birthweight less than the 10th percentile for the gestational age [18];-Fetuses large for gestational age—birthweight >90th percentile for gestational age [19];-Premature birth—any birth before 37 weeks of gestation [20];-PROM (prelabor rupture of membranes)—the rupture of membranes prior to the onset of labor [21];-Chorioamnionitis—infection of the amniotic sac and membranes [22];-Gestational hypertension—blood pressure above 140/90 mmHg after 20 weeks in previously normotensive women without proteinuria or other signs/symptoms of preeclampsia-related organ dysfunction [15];-Preeclampsia was defined as the new onset of hypertension and proteinuria or hypertension and significant end-organ dysfunction with/without proteinuria after 20 weeks of gestation in a previously normotensive woman/new onset of proteinuria, significant end-organ dysfunction or both after 20 weeks of gestation in a woman with chronic/pre-existing hypertension (it may also develop postpartum) [15].-HELLP syndrome was defined as the presence of hemolysis, elevated liver enzymes and low platelets [15];-Cholestasis of pregnancy—a liver disorder in the late second and early third trimester of pregnancy characterized by pruritus with increased serum bile acids and other liver function tests [23].

All patients signed the informed consent form, and we obtained approval from our hospital ethics committee to conduct this study (10,426/24.08.2021).

### Data Analysis

SPSS for Windows version 27.0 (IBM, Armonk, NY, USA) was used for data analyses. Kolmogorov–Smirnov, Shapiro–Wilk, Student’s *t*-test, and Mann–Whitney U were used were used for descriptive statistics. 

We performed a descriptive statistical analysis using the Statistical Package for the Social Sciences (version 29.0.0.0, IBM Corp., Armonk, NY, USA). Categorical variables were counted and expressed as frequencies. To check for significant differences between the two groups (adolescent underage pregnant women versus control group), the chi-square or Fisher’s test for categorical variables were used. A two-tailed *p*-value < 0.05 was considered significant. The risks were estimated using Fisher’s exact test with a *p*-value and 95% Confidence Interval (CI).

## 3. Results

During the study period, 2015–2021, of the 43,080 women who were admitted and gave birth in the Iasi “Cuza Voda” University Hospital, 41,998 (97.49%) were adults and 1082 (2.51%) were underage adolescents.

Figure 1 shows the trend of teenagers who gave birth during this 7-year study in our tertiary referral center, and Table 1 and Table 2 show their age distributions. Teenagers more frequently came from rural areas, and the difference was statically significant at *p* = 0.037.

There was a decline in the rate of teenage pregnant women from 2015 to 2019, with 2019 being the year with the lowest teenage pregnancy rate in our hospital (119 cases—11%). During the two pandemic years that followed, the rates of teenage pregnancies showed an upward trend compared to 2019.

There were 141 recurrent teenage pregnancies, with the majority of our minor patients being, as expected, primiparous. In Romania, teenage pregnant women who have no control of their fertility remain exposed to recurrent, closely spaced pregnancies and to all inherent unfavorable outcomes. In the present study, the percentage of multiparous teenagers was 9.1%, while 0.3% of the pregnant teenagers delivered three children between the age of 12 and 17 years.

The abortion rate in these cases was low at 46 cases, accounting for only 4.3% of all teenagers. In 60% of these cases, elective pregnancy termination was performed, with only 40% being spontaneous first trimester abortions or miscarriages.

### 3.1. Adverse Perinatal and Neonatal Outcomes in Teenage Pregnancies

#### 3.1.1. Fetal Malformations

We identified 25 cases of fetal malformations, representing 2.3% of the 1082 minor adolescents who gave birth in our hospital. The types of malformations detected are listed in Table 3.

#### 3.1.2. Twin Pregnancy

We identified 10 cases of twin pregnancy, representing 0.9% of our subpopulation group. The type of twin pregnancy and also the complications that occurred during the pregnancy are described in Table 4.

#### 3.1.3. Maternal Anemia

Maternal anemia was detected based on antepartum hemoglobin and hematocrit levels in 627 young patients, with a difference in hemoglobin and hematocrit levels between pregnant women with and without anemia exhibiting a *p* value of less than 0.001.

#### 3.1.4. Fetal Growth Pathology (FGR, LGA, SGA)

In total, 96 cases of fetal growth restriction (8.9%), 88 cases of fetuses that were small for gestational age (8.1%) and 28 cases of fetuses that were large for gestational age (2.6%) were detected. In establishing this diagnosis with accuracy, we also used the Ballard neonatal scores.

#### 3.1.5. Premature Birth and/or PROM

One hundred sixty of our patients gave birth prematurely, representing 14.8% of all teenage pregnant women, with forty-four (4.1%) of them at less than 32 weeks gestational age. We found no statistically significant difference between rural and urban residents in the percentage of premature births in teenage mothers (*p*—0.201). In six cases (0.6%), chorioamnionitis was detected (Table 5).

#### 3.1.6. Pregnancy Infections (Genital Tract Infections, Urinary Tract Infections and Chorioamnionitis)

Of the study’s pregnant teenagers, 55.5 % had no genital infection, and the remaining 45.5% had positive vaginal cultures. Candida was responsible for most genital infections and was observed in 223 pregnant teenagers (20.6%), followed by E. coli genital tract infections—143 patients (132%)—and Group B Streptococcus in only 24 patients (5%).

Only 69 patients had urinary tract infections, which account for 6.4% of all teenage pregnancies. The most commonly involved microorganisms were as expected: Escherichia coli (43 cases—4%) and Enterococcus (8 cases—0.7%).

#### 3.1.7. Hypertensive Disorders (Pregnancy Hypertension, Preeclampsia and HELLP Syndrome)

Of the pregnancy-associated conditions that we detected, 18 cases involved pregnancy hypertension (1.7%), 16 cases involved preeclampsia (1.5%) and 2 cases involved HELLP syndrome (0.2%).

#### 3.1.8. Other Conditions (Cholestasis of Pregnancy, Condylomatosis and Infection with Hepatitis B and HIV)

We found three cases of hepatitis B infection (0.3%), one case of HIV infection (0.1%) and eight cases of cholestasis of pregnancy (0.7%). We also noticed a high rate of condylomatosis in this young population, and it was statistically significant in more cases than in the adult pregnant women group with a *p* value less than 0.001.

When the outcomes and complications in the cohort of pregnant teenagers who gave birth in our hospital during the 7-year study were compared with the cohort of women over 18 years old that gave birth during the same interval in our hospital, we found some interesting results (Table 6).

We also noticed a high rate of condylomatosis in this young population, and it was statistically significant in more cases than in the adult pregnant women group with a *p* value less than 0.001.

We detected statistically significant higher rates of fetal malformation, premature birth, FGR and SGA fetal growth conditions, preeclampsia, condylomatosis and vaginal infection with E. coli in our cohort of teenagers (Table 6). In this subpopulation, the rate for premature birth at less than 32 weeks gestation was 3.26-fold higher (95% CI, 1.84–5.76, 95%), and for condylomatosis, it was 3.25-fold higher (95% CI, 1.80–5.89, 95%). These risks were estimated using the *p*-value from Fisher’s exact test.

## 4. Discussion

Teenage pregnancy is associated with many risks concerning both the life of the young mother and her baby. Pregnancy can negatively impact a variety of aspects of a young mother’s life, such as health, education and/or future employment perspectives [24]. Data from the literature reveal that teenage pregnancy poses high-risks [25] as a result of a combination of factors: the biological immaturity of teenagers, a lack of sexual education or lack of accessibility to it, poor perinatal care (mostly in low-income countries or lack of healthcare providers), inadequate maternal nutrition and last but not least stress with its multiple negative effects. Low literacy likewise comports the risk of unappropriated health decisions and access to available healthcare services, and it comprises well-recognized and continuous source of unfavorable maternal and fetal outcomes [26]. 

Teenage birth rates dropped to 7% in teenage girls aged between 15 and 17 years and dropped to 4% for older teenagers between 18 and 19 years in the United States. [27]. In Romania, where data on teenage births are incomplete, the incidence is around 3.7% according to Sedgh et al. [28].

In Romania, teenage pregnant women who have no control of their fertility remain exposed to recurrent, closely spaced pregnancies and to all inherent unfavorable outcomes.

### 4.1. Malformations

In our study, the rate of fetal congenital malformation among teenage mothers under 18 years of age was lower than in previous studies in the literature [29,30,31], although some studies report higher rates of fetal congenital malformations in very young (13–16 years) and young teenagers (17–18 years) [29,30]. Even though the studies offer conflicting results reporting higher incidences of gastroschisis and polydactyly or cardiac defects, we found that more than half of the cases had plurimalformative fetuses (14 cases of a total of 25) [31].

### 4.2. Twin Pregnancy

A study of twin birth rates in Europe by Heino et al. underlined the marked variations between countries, with a median twin birth rate of 1.68%. Our country has the lowest twin birth rate (0.9%) [32]. In our cohort of teenage women, the incidence of twin birth was 0.9%, which is similar to that reported in the above-mentioned study, and in our tertiary referral center, the incidence of twin pregnancy in adult population was 2.41%. The higher incidence of twin pregnancies and twin births in adult women could be explained by an increase in the number of pregnancies obtained by using assisted reproductive technology and the older age of first-time mothers [33].

### 4.3. Maternal Anemia

Maternal anemia seems to be the most-mentioned teenage pregnancy complication in the literature [34]. In our study, this condition was statistical insignificant, and it was slightly more prevalent in our teenage pregnancy group than in pregnant adult women (*p* = 0.222). According to some studies, the younger the teenage pregnant woman, the lower her hemoglobin level will be due to the necessary higher intake to assure the intense biological processes of both the young mother and the fetus [35,36].

Nevertheless, the high incidence of anemia detected in both groups underlines another serious medical problem that seems to be underestimated and poorly managed. More than 50% of our pregnant women in both our groups had anemia. The high percentage is close to the one detected by Ampiah et al. in their study in rural areas from Ghana [37]. Even though one might consider it to be less important than other conditions, anemia is a leading cause of maternal death and adverse pregnancy outcomes. Resulting from an association of multiple factors such as infections, nutrition deficiency and a lack of correct supplementation, anemia determines premature birth, low birth weight (FGR or SGA) and even low Apgar scores. Many studies underlined the correlation between maternal anemia and premature birth [15,16,17,18], and the correlation is a major source of increased neonatal morbidity and mortality. Moreover, anemia leading to FGR and SGA birth is involved in unfavorable fetal outcomes with high morbidity and mortality [17,18,36,38,39].

### 4.4. Fetal Growth Pathology

Many studies state that a growing organism does not efficiently mobilize fat reserves as an adult organism to adequately sustain fetal growth, generating impairments, and the pregnancy itself is affected by the mother’s competitive continued growth [40]. The data from our study are in agreement with those reported by Nkwabong et al. and Agbor et al., showing that the mean birth weight was smaller among teenagers than adult pregnant women, increasing with age [41,42]. Moreover, anemia might have a negative impact on fetal growth, causing both FGR or SGA birth, as stated before. A proper prenatal care with correct management and the mending of teenage mother’s anemia might reduce the number of preterm deliveries and FGR or SGA [17,36,37].

Moreover, our study identified a statistically significant lower proportion of LGA fetuses, and this is similar to other studies in the literature. As a possible explanation for these results, both Jain et al. and Karai et al. stated that very young mothers have lower obesity and lower glucose impairment and gestational diabetes incidence, all three of of which are involved in higher percentile fetal growth [31,43].

### 4.5. Preterm Birth and PROM 

The incidence of premature birth and especially of premature birth before 32 weeks gestation was significantly higher in our adolescent group compared with adult women. Premature birth is sub-classified into three groups: extremely preterm birth (<28 weeks gestation), very preterm birth (between 28 and 32 weeks of gestation) and late preterm birth (between 32 and 37 weeks of gestation). Preterm birth and PROM) share similar multifactorial etiopathogenesis. An incompletely developed cervix will be less resistant to clinical and subclinical infections. Infections initiate prostaglandin production, with prematurely ruptured membranes causing premature birth [43]. Moreover, the high incidences of deficient prenatal care with a delayed recognition of pregnancy and its possible complications are involved in both premature birth and PROM. Our results are similar with those reported by Fleming et al. and Jain et al. [43,44]. 

Premature birth continues to be a serious burden on our healthcare system, implying great costs and also being responsible for high morbidity and mortality, especially in middle-income countries such as Romania. Low birth weight and prematurity are responsible for the death of about 40% of the children under 5 years old worldwide [45].

The mean gestational age at childbirth in our underage adolescents was similar in both rural and urban areas (38.0 weeks of gestation in rural areas and 37.7 weeks in urban areas).

### 4.6. Maternal Infections

Underage pregnant adolescents are more prone toward contracting vaginal infections and sexually transmitted diseases. In our study, the analyzed subpopulation group presented a high incidence of E. coli vaginal infection, probably due to poor hygiene and health education. Vaginal infection was present in 44.5% of the teenagers in our study, statistically significantly higher than in the adult pregnant women group (*p* < 0.001) with respect to E. coli vaginal infection [1].

Although studies show that urinary infections are more frequent among pregnant teenagers, in our study only 6.4% of our patients had this associated pathology. As expected, based on our previous findings over half of these infections were due to E. coli. The rate is lower than that reported by Santos et al. in their study, and as to their frequency in underage adolescent pregnancies, it was less commonly encountered than anemia in our group of pregnant women [46].

Maternal vaginal and urinary infections are associated with premature birth and PROM, and these are well-known causes of maternal and fetal morbidity and mortality.

### 4.7. Hypertensive Disorders

Another spectrum of medical conditions that include a continuous and well-recognized cause of iatrogenic premature birth involves hypertensive disorders, including gestational hypertension, preeclampsia, eclampsia and HELLP syndrome. Amoadu et al. in their 2022 study reported high rates of hypertensive disorders and premature spontaneous and iatrogenic births in teenage pregnancies in both high and low-income countries [47,48]. A possible explanation for preterm births in pregnancies with hypertensive disorders might result from the consequent characteristic utero-placental ischemia [15,20,48].

Neal et al. state that this associated pathology is a major factor that strongly negatively impacts young maternal death, a finding that is also supported by the Finnish study by Leppalahti et al. 2013 [49,50]. Some other studies from Africa, Asia and South America that evaluated populations that were different from ours do not sustain this strong association between young adolescent pregnancy and adverse maternal outcomes due to racial and ethnic characteristics, such as the hypertensive disorder spectrum [6,10,15,26,47].

The number of preeclampsia cases was statistically significantly higher in our teenage cohort than in the adult pregnancy group (*p* < 0.001), whereas the rates of pregnancy hypertension and HELLP syndrome were statistically significantly higher in the adult women group. Although the percentage of patients with hypertensive disorders is higher in young nulliparous women, as expected, not all the studies in the literature agree, and their results are conflicting. A great number of studies support the observation that the hypertensive disorder spectrum complicates more teenage pregnancies than adult ones [31,33]. On the other hand, studies conducted in Oman and Canada found no statistically significant differences between young and adult mothers concerning the hypertensive disorder spectrum, and this might be due to sample sizes and social and biological risk factors [25,31,43].

### 4.8. Other Conditions (Cholestasis of Pregnancy, Condylomatosis, Hepatitis B and HIV Infection)

Of these diseases, only condylomatosis had a statistically significant higher rate in young adolescent mothers with a *p* value of under 0.001. Higher rates were also found only in the adult women group. 

The limitations of our study are the following: its retrospective character and the sample size that originates from a single tertiary center in our region. No data about the adolescents’ level of education, income, vitamin supplementation, nutrition and smoking or drug abuse habits were collected.

The strength of this research study is the large sample size obtained from the tertiary referral center database and the continuous verification for validity data, which mirrors obstetricians’ observations with respect to the effect of young maternal age on maternal and fetal outcomes in our region. Our results have markedly important implications, especially due to the fact that our country has incomplete data about this major health problem that has indisputable socioeconomic impacts on our health system.

## 5. Conclusions

Teenage pregnancies still remain a major health problem that burdens all countries worldwide regardless of their income. The problem requires solutions to prevent pregnancies in this young age segment and to improve both maternal and fetal outcomes. This particular stage in a woman’s development remains a critical one requiring more attention and sustained effort from parents, schools and governments. The higher the incidence of pregnancy in this population segment, the higher the need of a country or region to take measures to improve the quality of education and the healthcare system. 

Our study provides obstetricians with new insights into the particularities of underage adolescent pregnancy in our country and formulates directions for actions in order to improve our healthcare system. The major healthcare problems that we identified were as follows: a high incidence of anemia in our teenage pregnant women with its well-known complications and high incidences of premature birth, infections, preeclampsia and fetal growth pathology (SGA and FGR) in our young teenagers. All these conditions contribute to rising fetal and neonatal morbidity and mortality and all its related costs in our middle-income healthcare system. The rate of young adolescent pregnancies can be considered a reflection of the healthcare system in a country. Thus, in order improve the healthcare system, we need to focus on shifting from individual policies and actions to actions and policies that address both social and environmental structures. 

## Figures and Tables

**Figure 1 ijerph-20-01226-f001:**
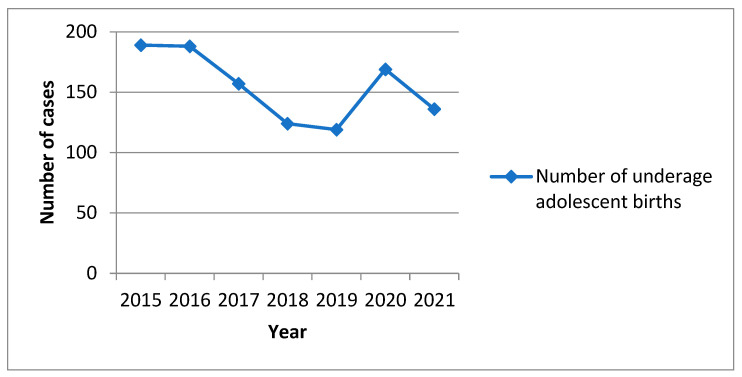
Trend in the number of underage adolescent pregnancy per year.

**Table 1 ijerph-20-01226-t001:** Characteristics of pregnant teenagers in our study population (part I).

	≤15 years					>15 Years			
	12 Years (n = 2)	13 Years (n = 12)	14 Years (n = 60)	15 Years (n = 205)	Total (n = 279)	16 Years (n = 425)	17 Years (n = 378)	Total (n = 803)	*p*-Value
Area									0.010 **
rural		9 (75.0%)	38 (63.3%)	158 (77.1%)	205 (73.5%)	346 (81.4%)	303 (80.2%)	649 (80.8%)	
urban	2 (100.0%)	3 (25.0%)	22 (36.7%)	47 (22.9%)	74 (26.5%)	79 (18.6%)	75 (19.8%)	154 (19.2%)	
Gestation									0.000 **
1	2 (100.0%)	12 (100.0%)	60 (100.0%)	189 (92.2%)	263 (94.3%)	372 (87.5%)	306 (81.0%)	678 (84.4%)	
2				16 (7.8%)	16 (5.7%)	49 (11.5%)	66 (17.5%)	115 (14.3%)	
3						4 (0.9%)	6 (1.6%)	10 (1.2%)	
Parity									0.000 **
1	2 (100.0%)	12 (100.0%)	60 (100.0%)	198 (96.6%)	272 (97.5%)	387 (91.1%)	325 (86.0%)	712 (88.7%)	
2				7 (3.4%)	7 (2.5%)	38 (8.9%)	50 (13.2%)	88 (11.0%)	
3							3 (0.8%)	3 (0.4%)	
Abortion									0.397
0	2 (100.0%)	12 (100.0%)	60 (100.0%)	196 (95.6%)	270 (96.8%)	409 (96.2%)	357 (94.4%)	766 (95.4%)	
1				9 (4.4%)	9 (3.2%)	13 (3.1%)	20 (5.3%)	33 (4.1%)	
2						3 (0.7%)	1 (0.3%)	4 (0.5%)	
Fetal malformations									0.836
No	2 (100.0%)	12 (100.0%)	59 (98.3%)	200 (97.6%)	273 (97.8%)	415 (97.6%)	369 (97.6%)	784 (97.6%)	
Yes			1 (1.7%)	5 (2.4%)	6 (2.2%)	10 (2.4%)	9 (2.4%)	19 (2.4%)	
Twin pregnancy									0.292
No	2 (100.0%)	11 (91.7%)	59 (98.3%)	203 (99.0%)	275 (98.6%)	420 (98.8%)	377 (99.7%)	797 (99.3%)	
Yes		1 (8.3%)	1 (1.7%)	2 (1.0%)	4 (1.4%)	5 (1.2%)	1 (0.3%)	6 (0.7%)	
Birth weight (g)	3235.00 ± 7.071	2740.00 ± 565.615	2940.51 ± 551.501	3043.56 ± 545.493	3010.58 ± 548.276	3035.21 ± 577.081	3110.82 ± 521.444	3071.03 ± 552.377	0.112 §
Birth weight (g)									0.120
AGA (acording to gestational age)	2 (100.0%)	9 (75.0%)	44 (73.3%)	163 (79.5%)	218 (78.1%)	349 (82.1%)	303 (80.2%)	652 (81.2%)	
SGA (small for gestational age)		2 (16.7%)	7 (11.7%)	23 (11.2%)	32 (11.5%)	28 (6.6%)	28 (7.4%)	56 (7.0%)	
FGR (fetal growth restrictiom)		1 (8.3%)	9 (15.0%)	13 (6.3%)	23 (8.2%)	37 (8.7%)	36 (9.5%)	73 (9.1%)	
LAGA (large for gestatinal age)				6 (2.9%)	6 (2.2%)	11 (2.6%)	11 (2.9%)	22 (2.7%)	

Pearson’s chi-squared test; ** *p* < 0.01 highly statistically significant; § Mann–Whitney U test.

**Table 2 ijerph-20-01226-t002:** Characteristics of pregnant teenagers in our study population (part II).

	≤15 Years					>15 Years			
	12 Years (n = 2)	13 Years (n = 12)	14 Years (n = 60)	15 Years (n = 205)	Total (n = 279)	16 Years (n = 425)	17 Years (n = 378)	Total (n = 803)	*p*-Value
Anemia									0.146
No		6 (50.0%)	22 (36.7%)	79 (38.5%)	107 (38.4%)	177 (41.6%)	171 (45.2%)	348 (43.3%)	
Yes	2 (100.0%)	6 (50.0%)	38 (63.3%)	126 (61.5%)	172 (61.6%)	248 (58.4%)	207 (54.8%)	455 (56.7%)	
Hb value (patients with anemia)	10.950 ± 1.344	11.200 ± 0.805	11.145 ± 0.917	11.157 ± 1.009	11.154 ± 0.978	10.973 ± 1.159	11.152 ± 0.921	11.054 ± 1.060	0.332 §
Hematocrit value (patients with anemia)	34.200 ± 2.263	33.133 ± 2.643	33.800 ± 3.184	33.487 ± 3.191	33.552 ± 3.146	33.016 ± 3.326	33.558 ± 2.984	33.262 ± 3.183	0.220 §
Gestational age at birth									0.344
24–28 weeks		1 (8.3%)		3 (1.5%)	4 (1.4%)	7 (1.6%)	2 (0.5%)	9 (1.1%)	
29–32 weeks			4 (6.7%)	8 (3.9%)	12 (4.3%)	13 (3.1%)	6 (1.6%)	19 (2.4%)	
33–36 weeks		4 (33.3%)	5 (8.3%)	23 (11.2%)	32 (11.5%)	51 (12.0%)	33 (8.7%)	84 (10.5%)	
≥37 weeks	2(100.0%)	7 (58.3%)	51 (85.0%)	171 (83.4%)	231 (82.8%)	354 (83.3%)	337 (89.2%)	691 (86.1%)	
Geastational age at birth	38.50 ± 0.707	36.17 ± 4.509	38.05 ± 2.561	37.93 ± 2.423	37.88 ± 2.578	37.82 ± 2.488	38.24 ± 1.967	38.01 ± 2.266	0.907 §
Chorioamniotitis									
No	2 (100.0%)	12 (100.0%)	58 (96.7%)	200 (97.6%)	272 (97.5%)	411 (96.7%)	369 (97.6%)	780 (97.1%)	0.585
Yes			1 (1.7%)		1 (0.4%)	4 (0.9%)	1 (0.3%)	5 (0.6%)	
Vaginal infection									0.579
No		9 (75.0%)	32 (53.3%)	110 (53.7%)	151 (54.1%)	231 (54.4%)	219 (57.9%)	450 (56.0%)	
Yes, from which:	2(100.0%)	3 (25.0%)	28 (46.7%)	95 (46.3%)	128 (45.9%)	194 (45.6%)	159 (42.1%)	353 (44.0%)	
Candida	1 (50.0%)	2 (66.7%)	14 (50.0%)	50 (52.6%)	67 (52.3%)	104 (53.6%)	98 (61.6%)	202 (57.2%)	0.341
*Citrobacter koseri*						2 (1.0%)	1 (0.6%)	3 (0.8%)	0.569
*Escherichia coli (E. coli)*	1 (50.0%)	2 (66.7%)	13 (46.4%)	35 (36.8%)	51 (39.8%)	76 (39.2%)	58 (36.5%)	134(38.0%)	0.708
*Gardnerella vaginalis*						1 (0.5%)	2 (1.3%)	3 (0.8%)	0.569
*Klebsiella pneumoniae*			1 (3.6%)	8 (8.4%)	9 (7.0%)	15 (7.7%)	4 (2.5%)	19 (5.4%)	0.495
*Enterococcus*			2 (7.1%)		2 (1.6%)	7 (3.6%)	5 (3.1%)	12 (3.4%)	0.372
*Enterobacter*				3 (3.2%)	3 (2.3%)	4 (2.1%)	2 (1.3%)	6 (1.7%)	0.705
*Group B Streptococcus*			1 (3.6%)	10 (10.5%)	11 (8.6%)	9 (4.6%)	4 (2.5%)	13 (3.7%)	0.029 *
*Pseudomonas aeruginosa*							1 (0.6%)	1 (0.3%)	1.000
*Proteus mirabilis*				1 (1.1%)	1 (0.8%)	1 (0.5%)	1 (0.6%)	2 (0.6%)	1.000
*S. aureus*							2 (1.3%)	2 (0.6%)	1.000
*Serratia marcescens*				1 (1.1%)	1 (0.8%)				0.266
Urinary infection									0.731
No	2 (100.0%)	12 (100.0%)	58 (96.7%)	188 (91.7%)	260 (93.2%)	398 (93.6%)	355 (93.9%)	753 (93.8%)	
Yes, from which:			2 (3.3%)	17 (8.3%)	19 (6.8%)	27 (6.4%)	23 (6.1%)	50 (6.2%)	
*E.* *coli*			2 (100.0%)	12 (70.6%)	14 (73.7%)	16 (59.3%)	15 (65.2%)	31 (62.0%)	0.363
*Klebsiella pneumoniae*						2 (7.4%)	4 (17.4%)	6 (12.0%)	0.177
*Enterococcus*				3 (17.6%)	3 (15.8%)	3 (11.1%)	3 (13.0%)	6 (12.0%)	0.699
*S. aureus*				2 (11.8%)	2 (10.5%)	3 (11.1%)	1 (4.3%)	4 (8.0%)	0.664
*Streptococcus agalactiae*				2 (11.8%)	2 (10.5%)	4 (14.8%)		4 (8.0%)	0.664
Pregnancy hypertension									0.667
No	2 (100.0%)	11 (91.7%)	59 (98.3%)	197 (96.1%)	269 (96.4%)	410 (96.5%)	367 (97.1%)	777 (96.8%)	
Yes		1 (8.3%)	1 (1.7%)	4 (2.0%)	6 (2.2%)	7 (1.6%)	5 (1.3%)	12 (1.5%)	
Preeclampsia				3 (1.5%)	3 (1.1%)	7 (1.6%)	6 (1.6%)	13 (1.6%)	
HELLP syndrome				1 (0.5%)	1 (0.4%)	1 (0.2%)		1 (0.1%)	
Other pathologies									
Chondylomatosis				2 (1.0%)	2 (0.7%)	6 (1.4%)	4 (1.1%)	10 (1.2%)	
Hepatitis B			1 (1.7%)	1 (0.5%)	2 (0.7%)		1 (0.3%)	1 (0.1%)	
Human immunodeficiency virus (HIV)						1 (0.2%)		1 (0.1%)	
Cholestasis of pregnancy				2 (1.0%)	2 (0.7%)	3 (0.7%)	3 (0.8%)	6 (0.7%)	

Pearson’s chi-squared test; * *p* < 0.05 statistically significant; § Mann–Whitney U test.

**Table 3 ijerph-20-01226-t003:** Types of fetal malformations encountered.

Type of Fetal Malformation	Number of Affected Fetuses
Cardiac defects	3 (0.27%)
Congenital malformations of the digestive tract	4 (0.36%)
Plurimalformative syndrome	14 (1.29%)
Central nervous system malformations	2 (0.18%)
Genital malformations	1 (0.09%)

**Table 4 ijerph-20-01226-t004:** Types and complications of twin pregnancies.

Twin Pregnancy	n
Twin pregnancy (with single fetal demise in the first trimester)	1 (0.09%)
Dichorionic diamniotic twin pregnancy	4 (0.36%)
Dichorionic diamniotic twin pregnancy, selective growth restriction	1 (0.09%)
Monochorionic monoamniotic twin pregnancy	1 (0.09%)
Monochorionic biamniotic twin pregnancy	2 (0.18%)
Monochorionic biamniotic twin pregnancy—twin-to-twin transfusion syndrome	1

**Table 5 ijerph-20-01226-t005:** Number of preterm births and their gestational age.

		n	%
Gestational age	24–28 weeks	13	1.2
	29–32 weeks	31	2.9
	33–36 weeks	116	10.7
	≥37 weeks	922	85.2
	Total	1082	100.0

**Table 6 ijerph-20-01226-t006:** Comparison between underage adolescents and pregnant women over 18 years of age concerning the the associated pathology.

Demographic Characteristic	Underage Teenagers (Frequency)	Adult Women > 18 Years Old (Frequency)	*p*-Value
Fetal malformation	2.31	1.09	0.000
Premature birth	14.78	8.88	0.000
Premature birth < 32 weeks	1.20	0.37	0.000
Chorioamniotitis	0.55	1.29	0.032
Twin pregnancy	0.92	2.41	0.001
Maternal anemia	57.94	56.08	0.222
FGR and SGA	17	9.31	0.000
LGA	2.58	4.04	0.015
Gestational hypertension	1.66	3.94	0.000
Preeclampsia	1.47	0.64	0.000
HELLP syndrome	0.18	0.36	0.445
Vaginal GBS infection	2.21	2.71	0.318
Vaginal Escherichia coli (*E. coli*) infection	17.09	10.03	0.000
Vaginal Candida infection	24.86	39.39	0.000
Urinary tract E. coli infection	4.15	12.74	0.000
Urinary tract Enterococus infection	0.83	4.15	0.000
Cholestasis of pregnancy	0.73	1.30	0.106
Hepatitis B	0.27	1.96	0.000
Condylomatosis	1.10	0.34	0.000
HIV	0.09	0.18	0.725

## Data Availability

The data used to support the findings of this study are available upon request.

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
