# Peer review of "Epidemiology of Adverse Outcomes in Teenage Pregnancy—A Northeastern Romanian Tertiary Referral Center"

_ijerph, 2023, doi:10.3390/ijerph20021226_

Round 1

Reviewer 1 Report

Authors of the article compared teenage maternal and their fetal outcomes regarding Romania where they have conducted a cohort study. A study was conducted in accordance with principles of the Declaration of Helsinki and Good Clinical Practice with informed consent obtained from all the members of the study group. Generally, you have highlighted an enormous problem appearing in Romania and how it influences the society. You have clearly emphasized inclusion & exclusion criteria for the study group. Unfortunately, I have the impression that you have conducted research which isn’t something innovative or a novel in today’s world.

I’d like to add some suggestions which will try to improve your work:

- you have prepared a very long introduction – it should be shortened to the essential information regarding your work.

- you didn’t mention the exact number of patients included into the study - there is only an information “all mothers aged less than 18 years, who delivered in our tertiary referral center, between January 2015 and December 2021”. It should be specified in abstract and manuscript - especially in lines 157-159.

- could you please explain why lines 375-383 are in italics?

- you have prepared 53 publications in the reference section - some of them are dated back almost to 1997… quite outdated. It would be better to remove outdated references and expand the reference list with publications dating only up to 5-6 years ago. It would improve the literature section concerning the work.

Reviewer 2 Report

The review of manuscript entitled „Incidence, Associated Factors and Impaired Outcomes in Un-2 derage Pregnancy - Disparities in a North-Eastern Romanian 3 Tertiary Referral Center”

Authors in their manuscript discuss very important problem of teenage maternal and their fetal outcomes. They perform  a retrospective cohort study of mothers aged less than 18 years and detected  significant higher rates of fetal malformation as well as premature birth, fetal growth restriction, SGA PE, and vaginal infection (E. coli) in teenagers. Also the rate for premature birth at less than 32 weeks gestation was much higher. They concluded, that pregnancy in adolescent women remains a major health problem and that it is necessary to prevent the pregnancy in young population.

        The authors observed an interesting trend of increasing pregnancies in adolescents during the COVID pandemic. In discussion they pointed, that teenage pregnancy is associated with many risks concerning both the life of the young mother and child as well as that pregnancy can negatively impact a variety of aspects of their life (health, education and future employment perspectives). In the discussion, the authors broadly discuss these problems resulting from the young age of the mother and concerning their health, but also the health of the children. They pointed, that  the proportion of teenage pregnancies is declining in Romania, as in other countries, such as the US.

My critical remarks:

1.      it was retrospective study with all limitation characteristic to this kind of studies

2.      only one center, tertiary but one

3.      more than 6,000 deliveries in one center is impressive, but raises questions about the quality of birth care

4.      in the results section the authors discuss the results and lead discussions, this should be done in the discussion section

5.      almost no results and observations regarding the health of women before pregnancy were shown, of course, that young age increases the risk of complications in the fetus and children born, but young women also get sick

6.      the discussion shows that the young age of the women was the main cause of fetal growth disorders and other pathologies ,what is the revelation in this statement?

The results of research conducted by colleagues are interesting, but only for doctors working in Romania. They are neither original nor revealing. They described facts that are generally known.

Round 2

Reviewer 1 Report

Authors have incorporated my suggestions and have modified the manuscript. I do not have any additional questions regarding the manuscript.